# Comparative Analysis of Emission Characteristics of In-Use China II–V Gasoline, Hybrid, Diesel-Fueled Vehicles

**Ye Zhang \*** , **Yating Song** , **Tianshi Feng** and **Yanyan Chen**

Beijing Engineering Research Center of Urban Transport Operation Guarantee, Beijing University of Technology, Beijing 100081, China
* Correspondence: zhangye@bjut.edu.cn

**Abstract:** Increasingly stringent regulations regarding vehicle emissions have contributed to the diversification of vehicle technologies, resulting in the increasing complexity of typical vehicles that make up a fleet. In order to investigate the real gas emissions of different typical vehicles, tests were conducted using a portable emission measurement system (PEMS) in Beijing and emission studies were conducted on eight light-duty passenger vehicles (LDPVs, including light-duty gasoline passenger vehicles and hybrid electric vehicles), eight heavy-duty passenger vehicles (HDPVs), and four light-duty trucks (LDTs). The results show that the emissions of relevant pollutants from LDPV meet the emission standard limits. The emission factors of $CO_2$, $CO$, $NO_X$, and HC of China IV and China V hybrid electric vehicles (HEVs) are much smaller than the emission standard limits and the emission factors of other vehicles, which have better emission reduction effects. Among LDPV, heavy-duty passenger vehicles (HDPVs), and LDT, the emissions of HDPV and LDT are extremely high. Emission characteristics vary on different types of roads, with the highest emission factors generally occurring on secondary roads. The micro-trip method was used to explore the influence of speed on emission factors. HEV are less sensitive to speed changes and can still maintain a low emission level at low speeds. The average speed and emission factors of HDPV in micro-trip has a strong correlation.

**Keywords:** real-world emissions; emission standards; emission factor; road attributes

## 1. Introduction

In recent years, the number of motor vehicles has increased worldwide. Road transport emissions are one of the primary causes of global warming, air pollution, and poor air quality [1]. In 2018, greenhouse gasses from the transportation sector accounted for 14% of global greenhouse gas emissions, of which road transport, passenger, and freight emissions accounted for 73% of the total sector emissions [2]. In addition, road traffic generates pollutants such as $CO$, $NO_X$, and HC. Among them, $CO$ is a serious hazard to the human body, $NO_X$ can damage the ozone layer, and HC is an important component of photochemical smog. According to the 2022 China Mobile Source Environmental Management Annual Report, in 2021, the total emission of four pollutants ($CO$, HC, $NO_X$, and PM) from motor vehicles in China was 15.577 million tons, of which the emissions of $CO$, HC, $NO_X$, and PM were 7.683 million tons, 2.004 million tons, 5.821 million tons, and 69,000 tons, respectively [3]. As of 2021, the number of motor vehicles in Beijing reached 6.8 million, which is a typical city in China with high motor vehicle ownership and serious environmental pollution [4].

In order to protect the environment and cope with climate change, Beijing has adopted a series of emission management measures. Analysis of emission characteristics is the basis for controlling urban motor vehicle emissions [5]. Motor vehicle emissions are influenced by a variety of factors, such as vehicle type (light-duty passenger vehicles, heavy-duty passenger vehicles, light-duty trucks, etc.) and operating conditions. Only by combining

local data can we better study motor vehicle emissions. For example, Wu et al. [6] calculated the emissions of major air pollutants in Beijing by using the data of motor vehicle ownership in Beijing from 2009 to 2019, combined with meteorological, driving mileage, vehicle speed, and other parameters, and distinguished the contribution and characteristics of emissions changes of different vehicles. Shen et al. [7] used PEMS in western Beijing by tracking the traffic flow on expressways and non-expressways to investigate the emission characteristics of particulate matter (PM) from light gasoline vehicles. Most of the existing studies have been speculations on emissions based on the data of vehicle ownership and macro traffic demand in Beijing, or the actual measurement data of emissions in a certain area and a certain road section in Beijing, and fewer of them have involved actual road test emission studies in a large area of Beijing.

This study was carried out on some roads in the area from the second ring to the sixth ring in Beijing, and the vehicle emission characteristics were studied based on PEMS to collect the emission information of $CO_2$, CO, $NO_X$, and HC for typical motor vehicles in Beijing. The focus was on comparing the differences in emissions of hybrid electric vehicles (HEVs), light-duty passenger vehicles (LDPVs), heavy-duty passenger vehicles (HDVs), and light-duty trucks (LDTs) under different traffic conditions. In addition, the emission characteristics of different vehicles under operating mode bins were also discussed.

## 2. Methods

### 2.1. Emission Measurement Systems

SENSORS-ECOSTAR PEMS was used to measure the emission of the test vehicle. The equipment was manufactured by SENSORS Company of the United States and met the measurement requirements of US CFR40 part 1065, UN-ECE R49, and EU582/2011. ECOSTAR is a new generation of modular equipment. The three core components are a flowmeter, a gas analyzer, and a particle analyzer. The flowmeter is used to monitor the instantaneous exhaust flow; the gas analyzer detects various gasses including CO, $CO_2$, NO, and $NO_2$; ECOSTAR also includes a global positioning system (GPS) and a meteorological center. The GPS receiver collects vehicle speed and position information every second and the meteorological center collects environmental parameters. During the test, the gas analyzer detects and collects the instantaneous data of $CO_2$ and pollutants at a time interval of 1 s. This instrument measured $CO_2$ and CO emissions through infrared absorption technology; $NO_X$ through ultraviolet absorption technology; and HC through a heated flame ionization detector. The hardware installation and software setup of the instrument were performed according to the instruction manual for each test.

### 2.2. Vehicle Information and Test Route

Motor vehicle road emission tests have been successively conducted in Beijing since 2014. Representative motor vehicles of different vehicle types in Beijing were recruited. Table 1 summarizes the specific information of the vehicles involved in the test, including eight LDPV, eight HDPV, and four LDT, giving a total of twenty vehicles. According to the three indicators of vehicle usage, emission standards, and power type, the 20 test vehicles were divided into 8 groups. According to China's Ministry of Ecology and Environment, in 2021, the CO, HC, $NO_X$, and PM of China II to China V standard vehicles accounted for 89.2%, 89.6%, 97.5%, and 96.4% of the total vehicle emissions, respectively [3]. Therefore, this dataset mainly included vehicles with emission standards from China II to China V. The cumulative mileage of vehicles ranged from 0.3 thousand km to 380 thousand km. During the test, all vehicles were empty. All gasoline vehicles and HEV are equipped with three-way catalyst (TWC) converters, and all diesel vehicles are equipped with an oxidation catalytic converter (DOC) and a catalytic diesel particulate filter (CDPF). Among them, the LDPV are designated vehicles for Beijing taxis and Shouqi taxi, the HDPV are the tour busses of China CYTS Tours Holding Co., Ltd., and the LDT are vehicles in the in-use industry, representing the typical vehicle technology for in-use vehicles in Beijing. This experiment only conducted the PEMS test under hot start. In order to reduce the impact of

driver characteristics on emissions, drivers with more than 5 years of driving experience were selected, and they were all skilled in driving the test vehicle. At the same time, it was ensured that the fuel of the test vehicle was filled at the gas station in Beijing, which met China V standards.

**Table 1.** Basic information of the test vehicle.

| Vehicle Category | Group | Vehicle Number | Emission Standards | Fuel | Displacements (L) | Mileage ($10^3$ km) | Model Year |
|---|---|---|---|---|---|---|---|
| LDPV | 1 | 2 | China II | gasoline | 1.6–2.4 | 26–38 | 2002/2010 |
| | 2 | 2 | China IV | gasoline | 1.6–2.4 | 26–38 | 2012 |
| | 3 | 2 | China IV | Gasoline-electric hybrid [1] | 1.5 | 2 | 2012 |
| | 4 | 2 | China V | Gasoline-electric hybrid [2] | 2.4 | 0.3 | 2014 |
| HDPV | 5 | 6 | China III | diesel | 8–8.5 | 65–70 | 2012 |
| | 6 | 2 | China IV | diesel | 8 | 58–60 | 2013 |
| LDT | 7 | 2 | China III | gasoline | 2.2 | 124–140 | 2007 |
| | 8 | 2 | China III | diesel | 4.8 | 70–167 | 2008 |

[1] Engine maximum net power: 73 kW; Electric powertrain maximum output power: 60 kW. [2] Engine maximum net power: 118 kW; Electric powertrain maximum output power: 105 kW.

The test route was the internal roads and ring roads in the area from the Second Ring Road to the Sixth Ring Road in Beijing, covering typical roads and many key points throughout Beijing as far as possible. The test route involved four types of roads: highways, expressways, main roads, and secondary roads, as shown in Figure 1. In Figure 1, the areas in different rings are distinguished by different colors. The inner ring is the area within the Second Ring Road, and the outwards areas are the third ring, the fourth ring, and the fifth ring, in that order. Each vehicle was tested once or twice. The test time of each vehicle ranged from 0.5 to 3 h, totaling 33 h, to ensure the authenticity and reliability of the collected data. Table A1 (Appendix A) gives the driving conditions parameters of each test vehicle.

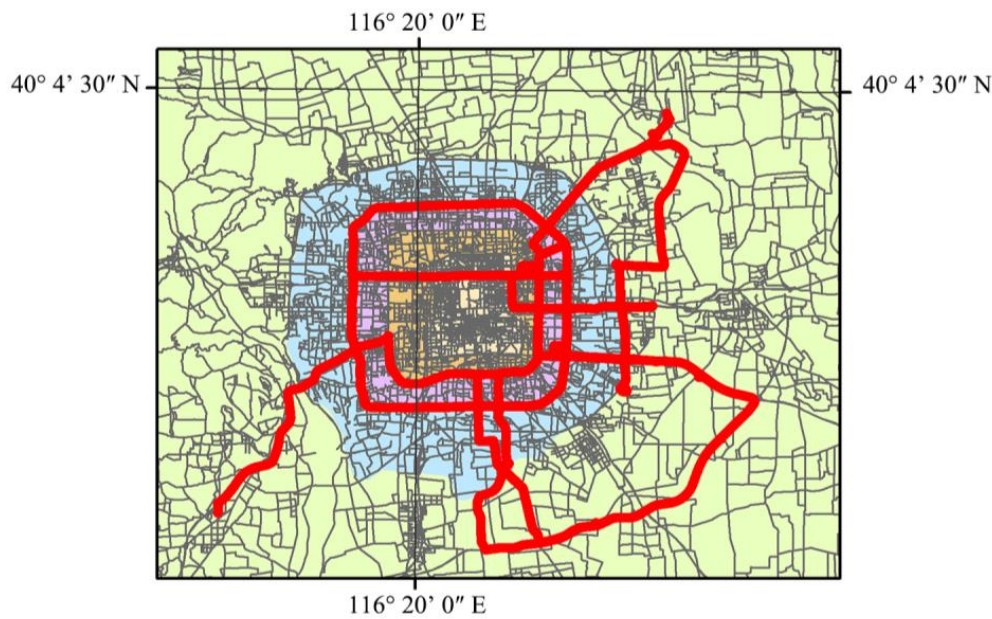

**Figure 1.** Test Route. The red line is the test route.

### 2.3. Data Processing

The emission factors for the test vehicle were calculated by Equation (1):

$$EF_i = \frac{3600 \sum_t ER_{i,t}}{\sum_t v_t} \tag{1}$$

where: $EF_i$ is the emission factor of pollutant $i$, g/km; $ER_{i,t}$ is the instantaneous emission rate of pollutant $i$ at second $t$, g/s; $v_t$ is the instantaneous speed of the vehicle at second $t$, km/h.

For HDPV, the influence of driving conditions on different routes is eliminated by Equation (2), and the emission factors of each vehicle are converted into the emission factor under the baseline traffic pattern.

$$EF_{0,i,n} = 3600 \frac{\sum_k \left( \overline{ER}_{i,k,n} \cdot P_{0,k} \right)}{V_0} \tag{2}$$

where $EF_{0,i,n}$ is the emission factor of pollutant $i$ in vehicle $n$ at the baseline traffic pattern, g/km; $\overline{ER}_{i,k,n}$ is the average emission rate of pollutant $i$ in vehicle $n$ under Bin $k$, g/s; $P_{0,k}$ is the time distribution ratio of the baseline traffic pattern in the Bin $k$; and $V_0$ is the average speed at the baseline traffic pattern, km/h. In this study, the baseline traffic pattern is taken as the average driving operating condition characteristics of all HDPV in the test. The operating mode bins adopt the same division form as Wang et al. [8], and a total of 22 operating mode bins are constructed. The VSP of HDPV is calculated by Equation (3) [9–11].

$$VSP = v(a + g \cdot \sin \alpha + 0.092) + 0.00021 \cdot v^3 \tag{3}$$

where $VSP$ is vehicle specific power, kw/h; $v$ is the vehicle instantaneous speed, m/s; $a$ is the vehicle instantaneous acceleration, m/s$^2$; $g$ is the gravitational acceleration, 9.81 m/s$^2$; $\alpha$ is the road slope, 0 for this study.

In order to refine the effect of each traffic flow segment on emissions while eliminating the effect of vehicle factors, the micro-trip method is used to divide the whole test condition. In this study, the whole test was divided into short-time continuous traffic flow with a time integration granularity of 300 s [12,13]. The relative emission factor was used to eliminate the influence of vehicle factors, and the relationship between the average speed of micro-trip and emissions was studied. The equation of relative emission factor is as follows.

$$REF_{i,n} = \frac{EF_{i,n}}{EF_{0,i,n}} \tag{4}$$

where: $REF_{i,n}$ is the relative emission factor of micro-trip for pollutant $i$ of vehicle $n$; $EF_{i,n}$ is the emission factor of that micro-trip, g/km; $EF_{0,i,n}$ is the emission factor of that test vehicle at the baseline traffic pattern, g/km.

## 3. Results and Discussion

### 3.1. Overview of Vehicle Emissions

The emission factors for each group of vehicles are summarized in Table 2. Figure 2 illustrates the differences in emission factors between the groups. Overall, HDPV and LDT emission factors were extremely high among all vehicles tested. For the emission factors of relevant pollutants in Group 1 compared with the China II standard, CO meets the emission standard, and the combined value of HC and NO$_X$ is 0.215 g/km, which is less than the emission limit value of 0.5 g/km. Compared with the China IV emission standard, the emission factors of test vehicles in Group 2 are less than the emission limit.

**Table 2.** $CO_2$, CO, $NO_X$, and HC emission factors for each group.

| Group | $CO_2$ (g/km) | CO (g/km) | $NO_X$ (g/km) | HC (g/km) |
|---|---|---|---|---|
| 1 | $249.46 \pm 99.60$ | $0.81 \pm 0.29$ | $0.165 \pm 0.04$ | $0.05 \pm 0.011$ |
| 2 | $270.28 \pm 151.021$ | $0.84 \pm 0.39$ | $0.04 \pm 0.011$ | $0.24 \pm 0.003$ |
| 3 | $109.548 \pm 2.36$ | $0.28 \pm 0.10$ | $0.0009 \pm 0.0007$ | $0.004 \pm 0.001$ |
| 4 | $98.72 \pm 4.61$ | $0.08 \pm 0.06$ | $0.011 \pm 0.001$ | $0.005 \pm 0.0002$ |
| 5 | $604.31 \pm 74.53$ | $5.80 \pm 2.09$ | $10.916 \pm 1.239$ | $0.22 \pm 0.06$ |
| 6 | $548.39 \pm 5.69$ | $5.70 \pm 0.16$ | $10.496 \pm 0.108$ | $0.129 \pm 0.03$ |
| 7 | $312.90 \pm 107.185$ | $18.917 \pm 9.74$ | $1.979 \pm 0.32$ | $2.50 \pm 0.36$ |
| 8 | $543.21 \pm 3.29$ | $2.21 \pm 0.67$ | $10.317 \pm 0.29$ | $0.38 \pm 0.192$ |

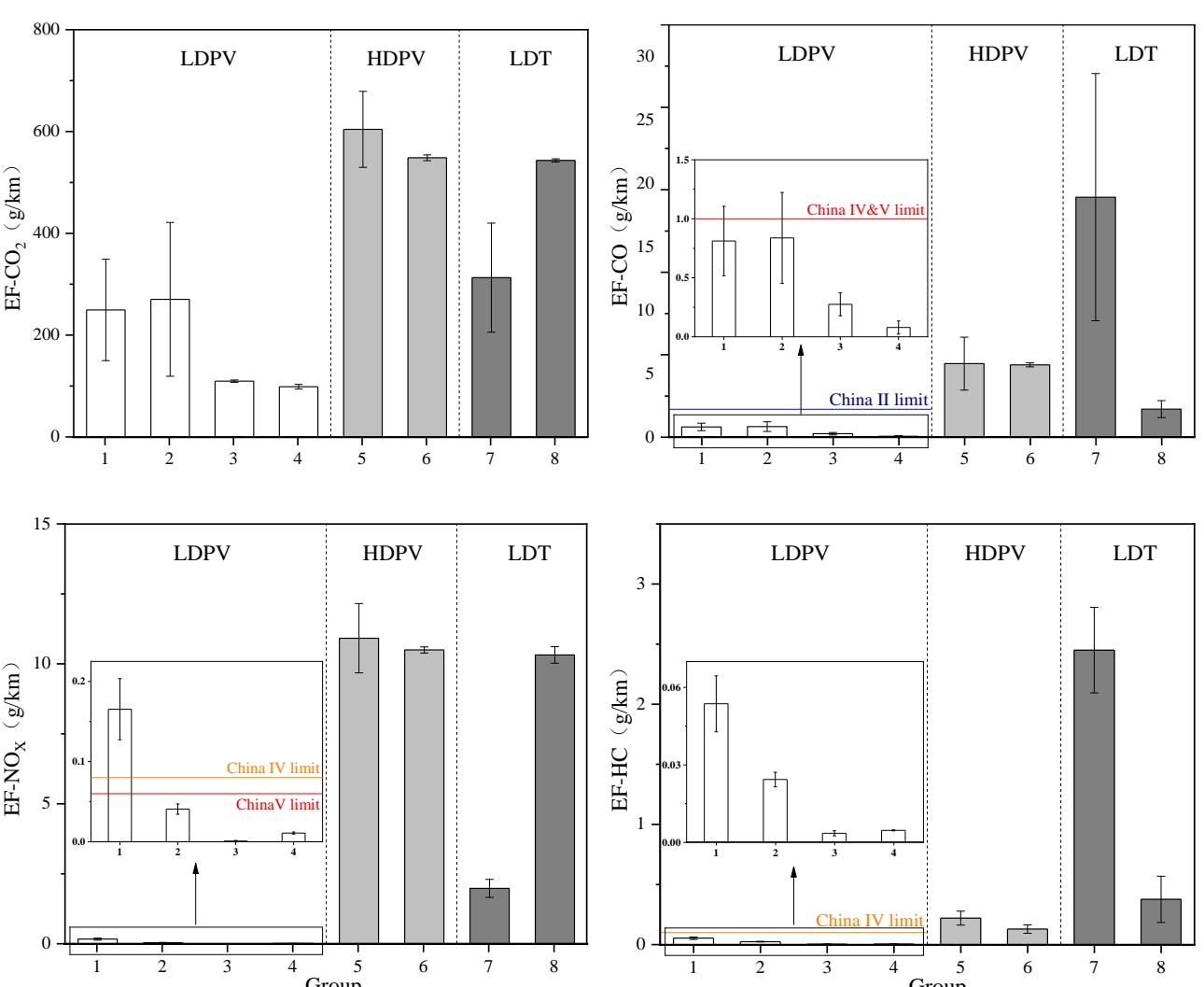

**Figure 2.** Emission factors of $CO_2$, CO, $NO_X$, and HC.

Both groups of HEV could meet the requirements of emission standards for CO and $NO_X$. Comparing Group 2 with Group 3, the emission factors of $CO_2$, CO, $NO_X$, and HC of the China IV HEV are far less than those of the gasoline vehicles under the same emission standards, which has an obvious emission reduction effect. Specifically, the differences in $CO_2$, CO, $NO_X$, and HC between the two groups are 71.6%, 43.3%, 99.3%, and 92.3%. In particular, the $NO_X$ emission factor of the China IV HEV is 0.0009 g/km, which is much smaller than the $NO_X$ emission factor of other vehicles. As can be seen from Table A1, the proportion of secondary roads in the test route for Group 3 is large. Compared with other types of roads, secondary roads have complex traffic conditions, as well as frequent

stops and starts. Frequent stops and starts result in a low fuel temperature, which is not conducive to the formation of $NO_X$ [14]. Due to the influence of vehicle weight on LDPV, although the emission standards of Group 4 of HEV (2100 kg) are higher than Group 3 (1805 kg), their emission factors of $NO_X$ and HC are still higher.

The emission factors of $CO_2$, CO, $NO_X$, and HC in Group 5 are higher than those in Group 6, and the emission factors of $CO_2$, CO, $NO_X$, and HC are reduced by 9.3%, 1.8%, 3.9%, and 41.4%, respectively.

The emission factors of CO, $NO_X$, and HC for light-duty gasoline trucks were ($18.917 \pm 2.21$), ($1.978 \pm 10.317$), and ($2.45 \pm 0.36$) g/km, respectively, which far exceeded the China III emission limits (CO = 5.22 g/km, $NO_X$ = 0.21 g/km, and HC = 0.29 g/km). In order to further reduce motor vehicle pollutant emissions, the Beijing Municipal People's Government issued a program to promote the phasing out and replacement of old motor vehicles with high emissions in 2020, accelerating the pace of phasing out and replacing China III emission standard gasoline cargo vehicles. Gasoline trucks have lower $CO_2$ and $NO_X$ emission factors and higher CO and HC emission factors compared to diesel trucks. This may be due to the combined effect of engine displacement and fuel type, etc. Diesel vehicles usually operate under fuel-scarce conditions, resulting in a reduction in CO emissions, and engine displacement has a positive effect on HC emission factors [15].

### 3.2. Emission Characteristics of Different Road Types

Figure 3a,b give the emission factors for each gaseous emission corresponding to the average speed within the micro-trip on different road types for HEV and LGPV. The micro-trip division of LDT was not considered in this study because the limited amount of data for LDT resulted in a small number of micro-trips on each type of road. Therefore, it was not sufficient to provide a correct description of the relationship between average speed and emission factors for LDT. As shown in the figure, the speed characteristics and emission characteristics of vehicles on different types of roads are different. The range of speed variation of vehicles on highways and expressways is large, and the driving speed on main roads and secondary roads is mostly concentrated below 40 km/h. The highest emission factors for each vehicle type generally appear on secondary roads, and a few of them appear in the low-speed section of the expressway. Vehicle traffic conditions and driving environments on different roads are different. In general, highways and expressways have flat roads and no signals; that is, the traffic conditions are relatively stable, so the overall vehicle driving speed is high. Main roads and secondary roads have more complex traffic conditions, a poorer driving environment, and more intersections and signals, resulting in frequent acceleration, deceleration, and idling, with speeds concentrated in the low speed range. It can be seen that the high emission factor is generally located in the interval where the average speed is low. It is expected that the emission factor will be very high at a very low speed (close to 0 km/h). It should be noted that the emission factor is also affected by accelerations, road grade, and vehicle factors. Therefore, the emission factor value in a short working condition is the result of many factors.

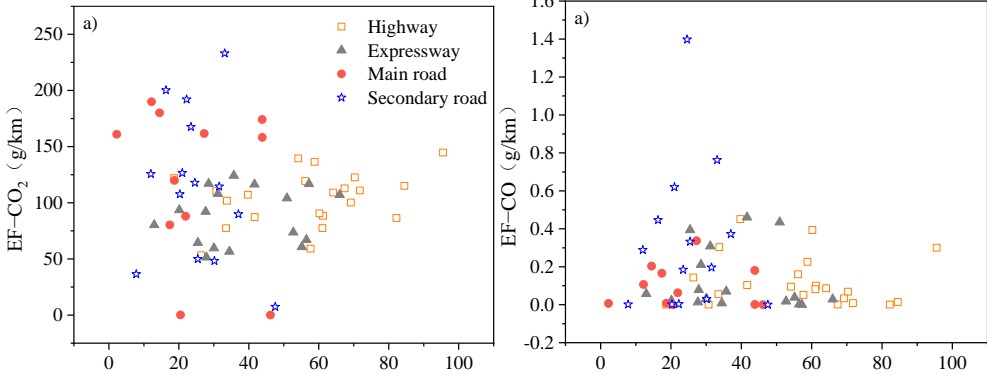

**Figure 3.** *Cont.*

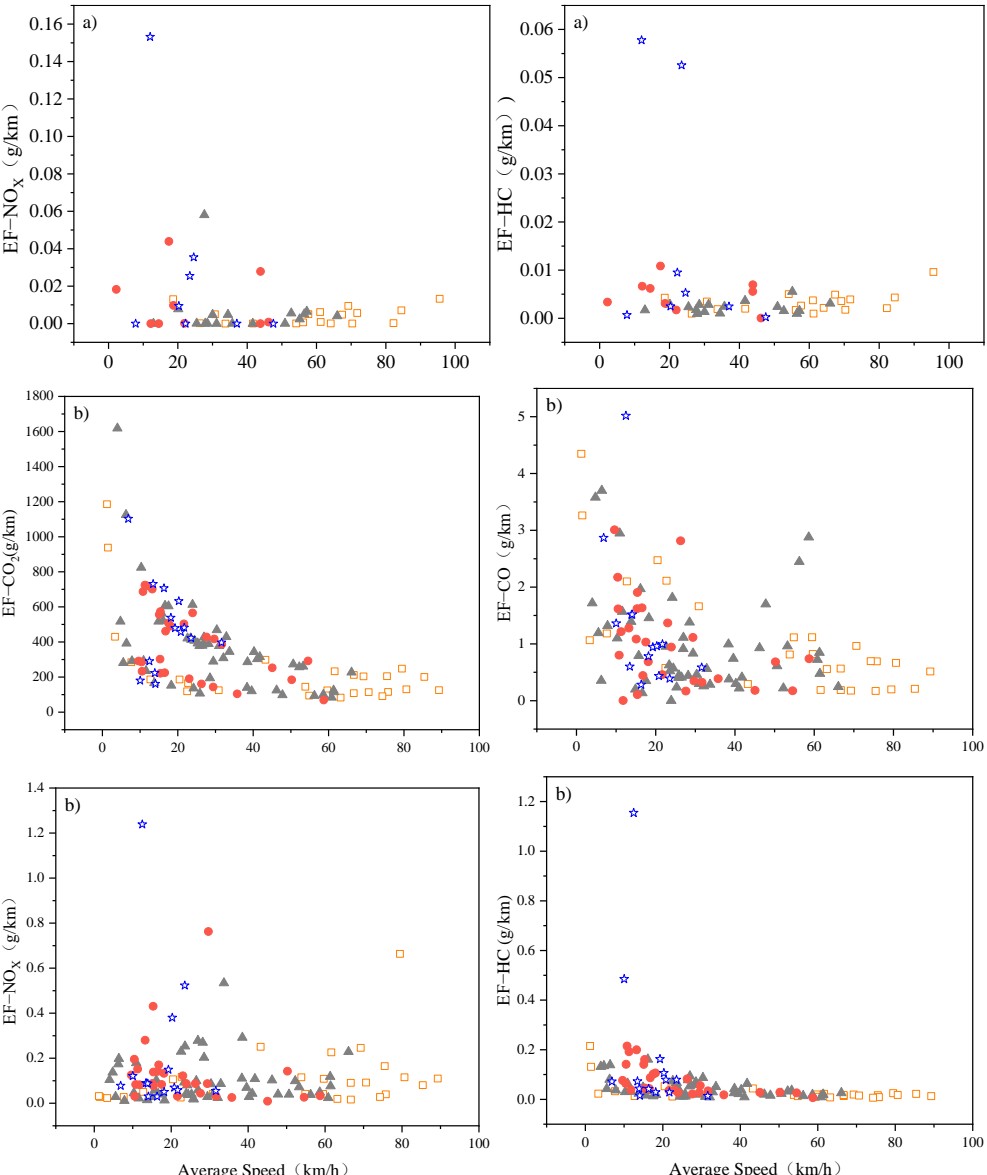

**Figure 3.** Emission factors corresponding to the average speed at micro-trip for different vehicle types. (**a**) Emission factors corresponding to the average speed of HEV at micro-trip; (**b**) Emission factor corresponding to the average speed of LGPV at micro-trip.

For HEV, the $CO_2$ and CO emission factors vary in a wide range with speed. $NO_X$ and HC emission factors are not sensitive to speed changes; 90% of the micro-trip $NO_X$ emission factors are distributed in 0–0.02 g/km and HC emission factors are distributed in 0–0.007 g/km. However, when compared with LGPV, the rate of change in emission factors for HEV is much smaller. In other words, the HEV is able to maintain a small emission even at lower speeds.

For LGPV, most of the micro-trips are distributed in low-speed intervals. When the average speed is below 15 km/h, the emission factors of $CO_2$ and CO significantly increase with the decrease in speed, and the emission factors of $NO_X$ and HC increase slightly with the decrease in speed. Overall, the long tail portion of an LGPV is mostly concentrated in highways and expressways, and each emission factor is low.

Figure 4 shows the relationship between the average vehicle speed and the relative emission factors of HDPV under micro-trips on different types of roads. From the figure, it can be seen that there is a strong correlation between $CO_2$, $NO_X$, HC relative emission factors, and average vehicle speed under micro-trip, while CO correlation is weaker. The

relationship between relative emission factors and average speed can be fitted by exponential, primary, and quadratic functions. The relationship between relative emission factors and the average speed of vehicles under micro-trip slightly varies for different road types. The fitting degree of $CO_2$ is high, with $R^2$ above 0.7 for all four road types, indicating that the change in vehicle speed has a strong influence on $CO_2$ emission factors. The weak correlation between the velocity of CO and the relative emission factor indicates that velocity is not a key factor affecting CO emissions. The HC emission factors for each micro-trip greatly vary, with a minimum relative emission factor of 0.8 and a maximum relative emission factor of 359.1, a difference of more than 400 times. Therefore, consciously avoiding low-speed driving can significantly reduce HC emissions, and can also effectively reduce $CO_2$, CO, and $NO_X$ emissions.

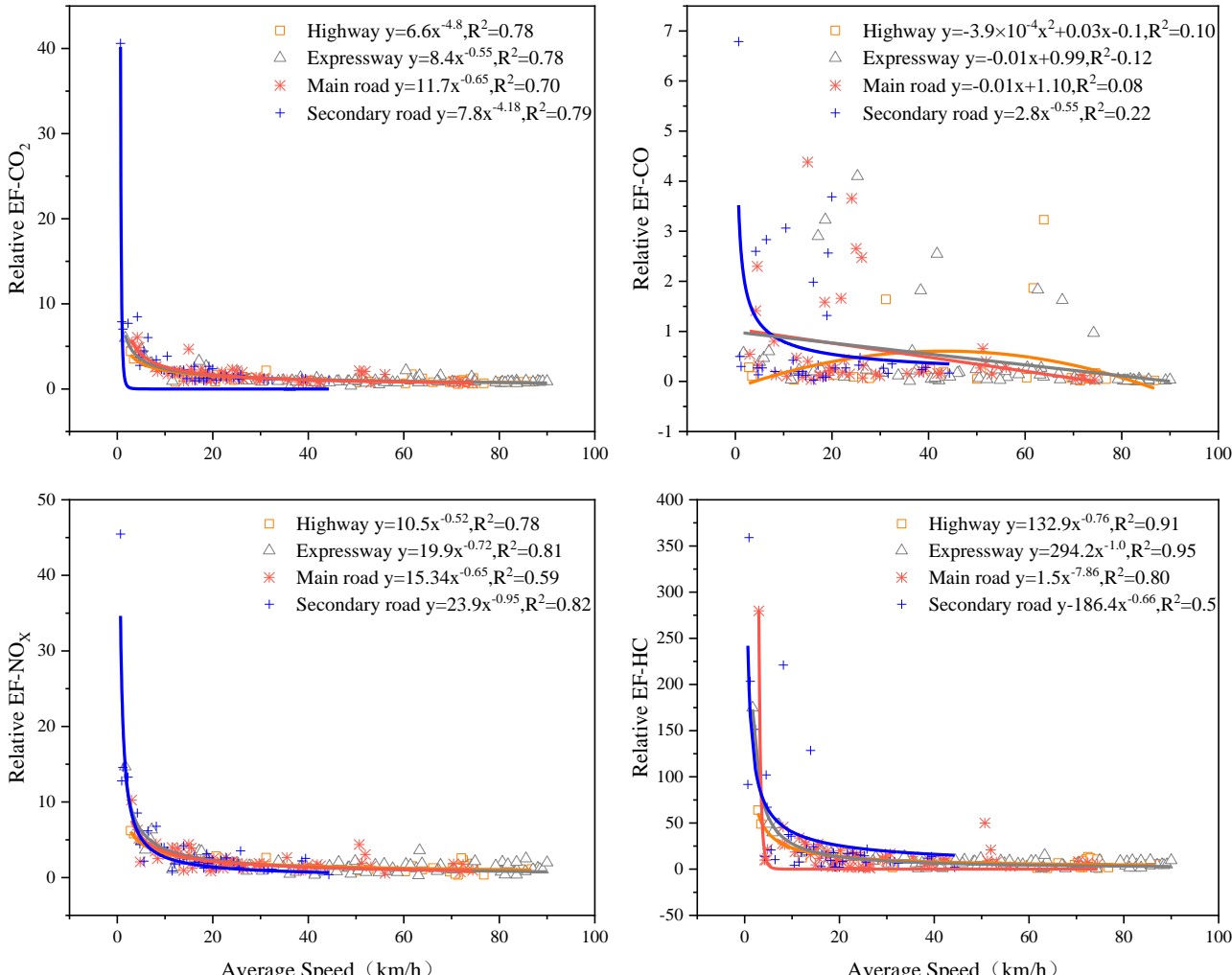

**Figure 4.** Correlations between relative emission factors of $CO_2$, CO, $NO_X$, and HC and average speed of micro-trips for HDPV.

Overall, the relative emission factors of $CO_2$, $NO_X$, and HC significantly increased when the average speed was below 15 km/h. When the average speed was between 40 km/h and 80 km/h, the relative emission factors of $CO_2$, $NO_X$, and HC were relatively stable with speed. Zhang et al. [12] studied the trend of $CO_2$ emissions from taxis on local roads and urban freeways in Beijing, and the results were similar to this study. However, the turning points of the relative emission factors of each emission gas were slightly different for different road types; for example, the relative emission factors of main roads showed a significant increase when the average speed was less than 5 km/h for HC.

## 4. Conclusions

In this study, PEMS was used to measure the emissions of $CO_2$, CO, $NO_X$, and HC from eight LDPV, eight HDPV, and four LDT in Beijing. The results show that HDPV and LDT have extremely high emissions among LDPV, HDPV, and LDT. LDPV can meet emission standards. Compared with other vehicle types, the emission factors of $CO_2$, CO, $NO_X$, and HC of HEV are significantly lower, with significant emission reduction benefits. With the upgrading of emission standards, the emission factors of $CO_2$, CO, $NO_X$, and HC of HDPV have all decreased. The emission factors of CO, $NO_X$, and HC of light-duty gasoline trucks are 18.917 g/km, 1.978 g/km, and 2.45 g/km, respectively, far exceeding the China III pollutant emission limits. Therefore, the pace of elimination of China III gasoline trucks in Beijing should be accelerated. Road attributes are one of the important factors affecting vehicle emissions. The speed of vehicles widely varies on highways and expressways, and the speed on main roads and secondary roads is mainly below 40 km/h. The highest value of emission factors generally occurs in the micro-trip with a low speed on secondary trunk roads. The exhaust emissions of HEV are less sensitive to speed changes and are able to maintain a small emission even at low speeds. For LGPV, when the speed is less than 15 km/h, with the decrease in speed, the emission factors of $CO_2$ and CO obviously increase, while the emission factors of $NO_X$ and HC slightly increase. When the speed is greater than 40 km/h, the emission factor is low. There is a strong correlation between the average speed and emission factor for HDPV in micro-trip. The variation in emission factors with speed is slightly different for different road types. Generally speaking, when the speed of the HDPV is less than 15 km/h, the emission factor significantly increases as the speed decreases. When the HDPV speed is from 40 km/h to 80 km/h, the emission factor is stably maintained at a low value.

**Author Contributions:** Conceptualization, Y.Z. and Y.S.; methodology, Y.Z. and Y.S.; software, Y.Z., Y.S. and T.F.; validation, T.F. and Y.C.; formal analysis, Y.C.; resources, Y.Z. and Y.C.; data curation, Y.Z., Y.S. and T.F.; writing—original draft preparation, Y.Z. and Y.S.; writing—review and editing, T.F. and Y.C.; supervision, Y.C.; project administration, Y.S. and Y.C.; funding acquisition, Y.Z. and Y.C. All authors have read and agreed to the published version of the manuscript.

**Funding:** This research was funded by Beijing Natural Science Foundation, grant number L201020.

**Data Availability Statement:** Data is contained within the article.

**Acknowledgments:** The authors would like to acknowledge the Beijing Natural Science Foundation.

**Conflicts of Interest:** The authors declare no conflict of interest.

## Appendix A

**Table A1.** Driving conditions of the test vehicle.

| Group | Test Vehicle | Distance (km) | Duration (min) | Highway | Expressway | Main Road | Secondary Road | Average Speed (km/h) | Ambient Temperature (°C) | RPA (m/s$^2$) | v·a$_{pos}$ [95] (m$^2$/s$^3$) |
|---|---|---|---|---|---|---|---|---|---|---|---|
| 1 | LGPV1 | 71 | 120 | 21.5% | 50.0% | 22.0% | 6.5% | 34.8 | 23–26 | 0.14 | 8.4 |
|   | LGPV2 | 68 | 160 | 3.7% | 63.6% | 22.7% | 10.0% | 22.0 | 23–26 | 0.16 | 8.6 |
| 2 | LGPV3 | 78 | 120 | 29.3% | 36.8% | 29.6% | 4.3% | 36.0 | 23–26 | 0.14 | 9.0 |
|   | LGPV4 | 55 | 120 | 49.1% | 38.0% | 6.0% | 6.0% | 25.6 | 23–26 | 0.13 | 7.7 |
| 3 | HEV1 | 58 | 155 | 36.7% | 41.7% | 11.2% | 10.3% | 36.4 | 23–26 | 0.15 | 10 |
|   | HEV2 | 22 | 40 | 36.2% | 0% | 14.1% | 49.7% | 34.0 | 23–26 | 0.22 | 15.8 |
| 4 | HEV3 | 48 | 70 | 41.8% | 34.3% | 17.6% | 6.3% | 41.5 | 23–26 | 0.13 | 11.0 |
|   | HEV4 | 86 | 110 | 31.9% | 41.6% | 10.8% | 15.6% | 45.0 | 23–26 | 0.13 | 13.2 |
| 5 | HDPV1 | 69 | 140 | 11.6% | 29.9% | 32.2% | 26.4% | 30.0 | 23–26 | 0.12 | 6.2 |
|   | HDPV2 | 86 | 140 | 13.6% | 38.8% | 28.6% | 19.0% | 36.2 | 23–26 | 0.13 | 8.4 |
|   | HDPV3 | 90 | 130 | 19.9% | 38.9% | 27.3% | 13.9% | 40.9 | 23–26 | 0.10 | 8.1 |
|   | HDPV4 | 81 | 120 | 44.8% | 19.7% | 27.8% | 7.7% | 44.2 | 23–26 | 0.12 | 8.4 |
|   | HDPV5 | 77 | 100 | 13.4% | 52.2% | 18.3% | 16.1% | 43.6 | 23–26 | 0.09 | 8.2 |
|   | HDPV6 | 50 | 55 | 27.0% | 45.1% | 11.1% | 16.8% | 57.5 | 23–26 | 0.07 | 7.4 |
| 6 | HDPV7 | 127 | 100 | 14.1% | 43.9% | 26.8% | 15.2% | 47.2 | 23–26 | 0.11 | 9.6 |
|   | HDPV8 | 79 | 90 | 14.1% | 46.3% | 14.5% | 25.1% | 46.4 | 23–26 | 0.11 | 9.9 |
| 7 | LDT1 | 68 | 160 | 2.0% | 34.9% | 45.9% | 17.2% | 25.0 | 10–13 | 0.12 | 5.7 |
|   | LDT2 | 69 | 100 | 0% | 82.1% | 5.9% | 12.0% | 40.8 | 10–13 | 0.12 | 6.2 |
| 8 | LDT3 | 37 | 65 | 20.3% | 22.7% | 14.6% | 42.5% | 34.0 | 10–13 | 0.12 | 7.8 |
|   | LDT4 | 47 | 115 | 42.5% | 45.6% | 2.1% | 9.7% | 24.3 | 10–13 | 0.14 | 6.1 |

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
