# Peer review of "Comparative Analysis of Emission Characteristics of In-Use China II–V Gasoline, Hybrid, Diesel-Fueled Vehicles"

_atmosphere, doi:10.3390/atmos14020272_

Round 1

Reviewer 1 Report

Review

The paper presents emission factors of gasoline hybrid and diesel passenger cars and trucks of various emission levels (China II-V). An analysis in function of speed and VSP is also presented. The results are useful but the manuscript needs some improvement.

Title

what “multi-vehicles” mean? Why not to clearly mention the vehicles tested? For example

Comparative analysis of emission characteristics of in-use China II-V gasoline electric hybrid, diesel passenger cars and trucks

Abstract:

The reader is confused with the terminology. The authors tested LDPVs, HDPVs and trucks. Then HEVs and LGPVs are mentioned, without clarifying to which category they belong.

I think it is worth mentioning that HDPVs and trucks had extremely high emissions in the order of g/km.

Introduction

The language needs improvement from a native English speaker. For example, first sentence:

Line 24: “In recent years, the global motor vehicle has increased significantly”

The authors probably mean “the number of motor vehicles globally has increased”

Lines 25-27: “Motor vehicle exhaust emissions have become a major source of urban pollution, posing a huge challenge to urban air quality management [1].

Citation [1] is a buildings ventilation study. A proper citation should be added that has analyzed the impact of vehicles on air pollution (e.g. as a percentage to compared to other sources of pollution)

Line35: what is vehicle type? Examples in brackets should be given

Line 36: “only combining local data” what does this mean? What kind of data need to be combined? If the authors refer to Wu et al., then the next sentence should begin as: “For example, Wu et al….”

Line 42: “particulate matter (PN)” either should be particle number (PN) or particulate matter (PM). Probably PN is the correct (based on the cited study)

Line 50: define HEV, LDPV, HDV

Line51: “traffic factors” should be replaced with “traffic conditions”

Methods

Lines 60-61. The PEMS could measure PN, but no results were presented. The authors should add them (eg in the Appendix)

PEMS: The authors should give a reference for the instrument they used.

Lines 68-70: “According to China's Ministry of Ecology and Environment, pollutant emissions from China II to China V standard vehicles will account for about 90% or more of total vehicle emissions in 2020[2].”

Could the authors give exact numbers (And not assumptions for 2020) as now we are in 2023?

Line 76: define CYT

Line 85: what does “giving full play to the role of liaison lines” mean?

The authors should give in the Appendix a table with the summary info of the test route for each vehicle including at least:

Ambient temperature, shares of urban rural, motorway (or any other classification of the authors), mean speed, total distance, duration. V x a, RPA (see RDE regulation, usually the PEMS gives this info)

What was the payload for LDPV and trucks when tested?

Each vehicle was tested once or more times?

Table 1: Why China II and IV were combined? Would it make sense to combine the two China II and the China IV instead ?

Why there is no mileage for some categories?

Why the trucks of category 7 are 2.2 L? Such engine displacement is usually for vans, not trucks.

Info about the hybrid vehicles should be given (eg electric motor power)

In general, is the model year of the vehicles available to be added?

Table 2: Is there any citation for the constant that were used either the patent in Chinese?

Line 136: The 300 s window was moving average or the trip was divided in 300 s bins? Why 300 s was selected and not a different duration?

I could not find citation [9]

Results

Lines 149-150 and Table 4. The literature review is very poor. There are many studies with PEMS and the selection of a few studies seems random. For example, only in “Atmosphere” journal a fast search reveals at least the following papers that should also be added if relevant:

Atmosphere 2019, 10, 535; doi:10.3390/atmos10090535Characterization of Real-World Pollutant Emissions and Fuel Consumption of Heavy-Duty Diesel Trucks with Latest Emissions Control

Atmosphere 2020, 11, 64; doi:10.3390/atmos11010064 Transient Characterization of Automotive Exhaust Emission from Different Vehicle Types Based on On-Road Measurements

Atmosphere 2020, 11, 765; doi:10.3390/atmos11070765 Capturing the Variability in Instantaneous Vehicle Emissions Based on Field Test Data

Atmosphere 2021, 12, 1125. doi: 10.3390/atmos12091125 Characterization of Exhaust CO, HC and NOx Emissions from Light-Duty Vehicles under Real Driving Conditions

Energies 2022, 15, 7886. doi: 10.3390/en15217886 Analysis of Exhaust Emissions from Heavy-Duty Vehicles on Different Applications

Energies 2022, 15, 8691. doi: 10.3390/en15228691Analysis of the Exhaust Emissions of Hybrid Vehicles for the Current and Future RDE Driving Cycle

Plus studies in the papers that the authors cited already in the table

Furthermore table 4 needs some corrections:

Huo: tested many emission levels (Euro 0 to IV). Why only Euro IV was considered and not Euro II?

May: The study has light duty and heavy duty vehicles. Why the heavy-duty were not considered in the table? The values are given per kg fuel and not per km How the authors did the conversion?

Valverde: They were Euro 6b cars. A note should clarify for which cycles the emissions in the table are given

Wu: THC is also available in the original publication

Freddy should be Rosero: CO is 4.52 not 4.25. The original publication gives also THC.

Xie: I have no access to this paper in Chinese, but the numbers do not match with those in the abstract

Grigoratos: They were Euro VI vehicles

I suggest to remove table 2 completely or move it in the Appendix if it is expanded with more studies. Preferably it is enough to discuss in the text any comparison with the studies of Table 4, than have the table (which should also have info about the vehicles, and the routes they were tested at).

Line 153: The authors use latin and Arabic (eg China 2 and China 6) in the text. Why?

Line 156: LGPV appears first time. HEV (line 168) as well. There should be a match with the definitions in the Methods section

Figure 2: Why the emissions are compared with China 5 and 6 and not China 2 and 4 as the emission standard of the cars?

Line 162: Although the dependency of emission in function of engine displacement makes sense, it is not true with the installation of aftertreatment that practically remove the pollutants. Thus I would expect any impact only on the CO2

Figure 3: I suggest some of the symbols to be open (no filling) for better visualization.

In the materials the authors should explain what are the differences between highway, expressway, main and secondary roads, and why differences are expected from the type of road (other than the mean speed and traffic)

Why for HEV the emissions are given in g/km while for the other categories as relative differences?

I think giving everything in g/km is better for the readers and future citations. Or at least the value that the emission were normalized to should be given in the legend.

It is expected that at very low speed (eg with a lot of idling) the emissions will seem very high (as the denominator is a very small number) (eg the points with speed close to 0). This should be mentioned in the text or could be avoided with longer micro trips.

Figure 4: why fittings are given for HDPVs and not the other categories?

Other studies have investigated the effect of the speed or VSP on emissions. Eg

Atmosphere 2019, 10, 535; doi:10.3390/atmos10090535

Atmosphere 2020, 11, 765; doi:10.3390/atmos11070765

Atmosphere 2021, 12, 1125. https://doi.org/10.3390/atmos12091125

However, speed is not the main parameter, but accelerations, road grade etc can also have an impact and should be commented in the text: Eg

Atmosphere 2021, 12(8), 1011; https://doi.org/10.3390/atmos12081011

Conclusions

Line 282: nobody expects that the vehicles would respect the China 6 limits, as they were China 2-5 vehicles. The authors should compare with the respective limits.

Line 285: The conclusion that HEVs are the lowest emitting is not fully correct. I do not disagree, but it could due to the gasoline technology (and not the hybridization). The first two categories were diesel. Thus it is very important to add that the HEVs were gasoline:

“At the same time, the gasoline HEV with lower emission standards can fully…”’

Line 287: “The upgrading of emission standards to the current stage is difficult to cause a significant decrease in emission factors.”

What does this mean?

Line 290: “LGPV and trucks are more sensitive to speed changes”

Where does this conclusion come from? They only that can be said is that the HEVs were not so sensitive to speeds changes (actually the authors mean there was no correlation of emissions with speed?)

Lines 295-297: I would delete this sentence. The fitting in function of speed was not so good, and it shows an increase at low speeds (which is logical). Furthermore, the results were not compared with other studies in the literature.

Reviewer 2 Report

Manuscript Number; Atmosphere-2131104

Title; Comparative analysis of emission characteristics of in-use hybrids and multi-vehicles

Although the topic is of interest to the Scientific community, before considering it for publication, this paper should be improved. Authors should reconsider the main objective of the paper according to the content. They should try to synthesize and emphasize the study's main findings and avoid long sentences. Furthermore, authors should avoid drawing risky conclusions.

Evaluation; Major Revision

1.    Keywords; Must to revised; spelling and avoiding general and plural terms and multiple concepts (avoid, for example, 'and', 'of').

Unsuitable >>> Portable emission measurement systems (PEMS)

     Hybrid electric vehicles (HEVs)

2.    Line 27-29; According to the 2021 China Mobile Source Environmental Management Annual Report, the total emissions of four pollutants (CO, 

HC, NOX, PM) from motor vehicles in China is 15.93 million tons [2].

It is unclear. Should be reported in each pollutant.

3.    Line 29-30; the number of motor vehicles in Beijing has reached 6.8 million.

What types of motor vehicles?  It should be a specific type of vehicle and fuel consumption use.

4.    Line 60-61; Particle analyzer can accurately detect the number concentration of particles  with particle size above 23 nm. It should be concern about UFPs that smaller than 23 nm. Because most study about UFPs suggest that motor vehicles are the main sources of UFPs.

Reference;

- Kwon, H.S.; Ryu, M.H.; Carlsten, C. Ultrafine particles:  Unique physicochemical properties relevant to health and disease.  Exp. Mol.  Med.  2020, 52,  318–328. 

- Phairuang, W., Amin, M., Hata, M., & Furuuchi, M. (2022). Airborne Nanoparticles (PM0. 1) in Southeast Asian Cities: A Review. Sustainability, 14(16), 10074.

- Schraufnagel,  D.E.  The health effects of ultrafine particles.  Exp.  Mol.  Med.  2020, 52, 311–317.

5.    Figure 1. Test Route. It is unclear. Don’t have any scale and grid line information.

6.    Please be careful about NOX. It should be a subscription (NOx) all of the main text.

7.    How about QA/QC in this study e.g. Data processing, etc.

8.    All of the main text, many numeric data are given with too many significant figures; 2 significant figures suffice, and 3 suffice in case the first significant figure is "1". 

E.g., line 105;  1.207 kg/m3.

 Line 191-192; The emission factors of CO and NOX for light-duty gasoline trucks are (18.917±2.213) and (1.978±10.317) g/km

9.    Figure 3. Emission factors, relative emission factors corresponding to the average speed at micro-trip for different vehicle types. a) Emission factors corresponding to the average speed of HEVs at micro-trip; b) Relative emission factor corresponding to the average speed of LGPV at micro-trip; c) 

Relative emission factor corresponding to the average speed of the truck at micro-trip.

Need to be revised. It is too small and unclear.

10. Conclusion; Please revise and combine them into only one paragraph in the conclusion. The conclusions could be further developed, there should be integrated data in the article.

Round 2

Reviewer 1 Report

The authors have improved the manuscript. A few things are not clear that need to be addressed:

Vehicle 7 with 2.2 l engine could be a van. Vehicle 8 with 4.8 l seems more heavy duty truck. Could the authors confirm the classification of the two vehicles as vans?

For the original version I asked if there is any other citation for the constant that were used for VSP calculation and not only the patent in Chinese. The authors removed the reference. Why? I asked if there is more support on the numbers that they used. Now there are no numbers and the reader does not know what values the authors were used.

PN: I am not sure why the data are not presented. If the PEMS is measuring PN >23 nm, then the results are in real time available. If the PEMS is measuring mass, then the authors should write PM (not PN). Then the integrated values could be given (as in Figure 2) without further analysis in function of speed. Or the authors should comment in the manuscript why they do not provide the data

What was the payload for LDPV and trucks when tested? The authors replied about the repetitions The question is was how much the vans and trucks and heavy-duty vehicles were loaded? In other words, were they empty or the authors added some weight?

Old citation [9}: The new link takes me to a page where user and password is needed. Is this reference necessary? It seems it’s not publicly available

I cannot find the previous results and figure in function of VSP. Why they were removed? Because in the experimental the VSP methodology is still described.

Table 3

The authors added some of the citations I suggested, but still many are missing, and there are many more from other journals (I recommended only a few from Atmosphere). As this paper does not need to be a review paper of other’s tests, please remove the table from the main text and have it as supplementary information if you see any value. Please add the values from your tests in the table

Reviewer 2 Report

This revised version is suitable for publication.

Author Response

Thank you so much for your review which gave us valuable suggestions! We deeply appreciate your recognition of our research work.